# Successive Waves of the COVID-19 Pandemic Had an Increasing Impact on Chronic Cardiovascular Patients in a Western Region of Romania

**DOI:** 10.3390/healthcare11081183

**Published:** 2023-04-20

**Authors:** Adelina Tudora, Diana Lungeanu, Adina Pop-Moldovan, Maria Puschita, Radu I. Lala

**Affiliations:** 1Arad County Clinical Emergency Hospital, 310037 Arad, Romania; 2Faculty of Medicine, “Vasile Goldis” Western University, 310025 Arad, Romania; 3Center for Modeling Biological Systems and Data Analysis, “Victor Babes” University of Medicine and Pharmacy, 300041 Timisoara, Romania; 4Department of Functional Sciences, Faculty of Medicine, “Victor Babes” University of Medicine and Pharmacy, 300041 Timisoara, Romania

**Keywords:** COVID-19, cardiovascular comorbidities, mortality, disease severity, pre-existing cardiovascular disease, SARS-CoV-2

## Abstract

Three years since the COVID-19 pandemic started, there is still little information about patients with chronic medical conditions, such as cardiovascular diseases (CVDs), who become infected with SARS-CoV-2. A retrospective analysis was performed to evaluate the impact of the COVID-19 pandemic on patients with cardiovascular comorbidities hospitalized with positive RT-PCR results for SARS-CoV-2 during the highest peaks of the first three pandemic waves: April 2020, October 2020, and November 2021. The primary outcome was in-hospital mortality; the secondary outcomes were length of hospitalization and required mechanical ventilation to assess the disease severity. Data were extracted from the hospital electronic database system: 680 eligible cases were identified out of 2919 patients. Mortality was the highest in wave 3 (31.9%) compared to the previous waves (13.6% and 25.8%). Hospitalization was also significantly longer in wave 3 (11.58 ± 5.34 vs. 8.94 ± 4.74 and 10.19 ± 5.06; *p* < 0.001), and so was the need for mechanical ventilation (21.7% vs. 8.2% and 9%; *p* < 0.001). Older age and male gender were confirmed as highly significant predictors of unfavorable outcomes. Ischemic heart disease worsened the odds of patients’ survival irrespective of the three pandemic waves (Breslow–Day test, *p* = 0.387), with a marginally significant Mantel–Haenszel common estimate for risk: OR = 1.604, 95% (0.996; 2.586). The significantly worse outcomes in wave 3 could have been influenced by a combination of factors: the low percentage of vaccinations in Romanian population, the more virulent delta strain, and pandemic attrition in the care provided to these patients with chronic CVDs.

## 1. Introduction

Three years since the start of the COVID-19 pandemic and with the emergence of anti-COVID-19 vaccines and antivirals, herd immunity has been achieved in most countries, with new subvariants of the virus causing mainly upper respiratory tract infections and low mortality rates. On the other hand, despite the determination of the medical community and health authorities to control the pandemic, there are still unanswered questions about the vulnerable or high-risk groups among patients with chronic medical conditions. Due to the virus binding to the angiotensin-converting enzyme 2 (ACE2) receptor, which is highly expressed in the lungs, heart, and vessels, the virus is now understood to have significant cardiovascular implications beyond respiratory tract infections [1,2,3]. The pathophysiology of most cardiovascular diseases (CVDs) involves the dysregulation of the renin-angiotensin system, which is further amplified by SARS-CoV-2 infection, so it is reasonable to look for an association between them. Early pandemic studies reported a high prevalence of cardiovascular comorbidities—hypertension (56%), obesity (41%), coronary artery disease (11%), and congestive heart failure (8%)—among patients infected with SARS-CoV-2 [4]. Moreover, in statistical models, COVID-19 was related to cardiovascular complications, such as cardiac injury, thrombotic and thromboembolic complications, and lipid and glucose metabolism dysregulation, which are partially explained by endothelial dysfunction, hypercoagulability, and systemic inflammatory responses [4,5,6]. Nevertheless, there have been some conflicting results; for example, some studies found that only patients with CVD who developed cardiac injury during SARS-CoV-2 infection were at high risk for adverse events [7], while others concluded that the CVD comorbidities had a modest effect on poor outcomes of COVID-19 (which was actually mainly dependent on age and gender) [8].

The worldwide COVID-19 vaccination drive that started at the beginning of 2021 first targeted the elderly and high-risk population (especially those with chronic CVDs) and might have favorably influenced the outcomes during the subsequent COVID-19 waves; therefore, conflicting results have been reported in this population [9,10]. Clinical studies have focused on long-term cardiovascular consequences after COVID-19; for example, Xie et al. [11] reported an increased risk for cardiovascular burden that included major adverse cardiovascular events (MACEs—myocardial infarction, stroke, and all-cause mortality), hypertension, cerebrovascular disease, myocarditis, heart failure, arrhythmias, and ischemic heart disease (IHD) one year after COVID-19.

We conducted a retrospective analysis to evaluate the impact of the COVID-19 pandemic on patients with cardiovascular comorbidities during the first three successive waves, when the risk of SARS-CoV-2 infection was at its highest and preventive measures were stretched to their limits, as determined by national statistics [12].

The primary objective was to assess the in-hospital mortality of these COVID-19 patients. The secondary objective was to evaluate the disease severity among them, assessed by the need for mechanical ventilation and length of hospitalization.

## 2. Patients and Methods

### 2.1. Study Design and Participants

This retrospective, cross-sectional study was conducted at a tertiary hospital in the western part of Romania, Arad County Clinical Emergency Hospital, during the peak waves of COVID-19 pandemic in this country: 1–30 April 2020, 1–31 October 2020, and 1–30 November 2021. Data were extracted from the hospital electronic database system and included all hospitalized patients with a positive reverse transcription polymerase chain reaction (RT-PCR) for SARS-C-V-2 and at least one cardiovascular comorbidity, namely hypertension, IHD (comprising chronic coronary syndrome, old myocardial infarction, acute coronary syndrome, history of coronary aortic by-pass, or history of stent implantation), myocardial infarction, heart failure, atrial fibrillation, valvular heart disease, peripheral artery disease, myocarditis, pulmonary embolism, and stroke. There were no exclusion criteria, and all data records had information concerning the primary and secondary objectives of the study. Of 2919 patients admitted to the hospital during these three periods, a total of 680 eligible cases met the inclusion criteria.

Figure 1 shows the study workflow.

### 2.2. Study Variables and Outcomes

The study’s primary outcome was in-hospital mortality at any time from admission to discharge. The secondary outcomes referred to COVID-19 severity assessed by the need for mechanical ventilation and the length of hospitalization.

The following variables were also considered: demographic data (age and sex), CVDs, associated comorbidities (obesity, chronic obstructive pulmonary disease, diabetes, chronic kidney disease), paraclinical variables (laboratory parameters and X-ray findings), noninvasive ventilation, mechanical ventilation, clinical variables (i.e., symptomatology), vaccination status in wave 3, and having received treatment for COVID-19.

### 2.3. Statistical Analysis

Descriptive statistics included the observed frequency counts (percentages) for categorical variables and the mean ± standard deviation or median (interquartile range) with Tukey’s hinges for numerical variables. Univariate non-parametric statistical tests were applied to compare the distribution of numerical data across two or multiple groups, as appropriate (either the Mann–Whitney U test or the Kruskal–Wallis test, respectively). The chi-square statistical test (either asymptotic, Fisher’s exact test, or Monte-Carlo simulation with 10,000 samples) was applied to check the statistical significance of the associations between the categorical variables.

The association of cardiovascular diseases with fatal outcomes was investigated based on the odds ratio (OR) values, calculated for each of the three wave groups. The Breslow–Day test was applied for the homogeneity of ORs across the strata, and the Mantel–Haenszel test was used for conditional independence of the disease and mortality. For medically meaningful and significant associations, the Mantel–Haenszel summary adjusted OR value was calculated, with the corresponding 95% confidence intervals.

The statistical analysis was conducted at a 95% level of confidence and a 5% level of statistical significance. All reported probability values were two-tailed.

Statistical analysis was performed with IBM SPSS statistical software, version 20, and R software, version 4.0.5.

## 3. Results

### 3.1. Patients’ Characteristics and Study Outcomes

The 680 patients were 66 ± 11 year old; 48% were male and 52% female; and the overall mortality was greater than 25%. Table 1 presents the descriptive statistics for the patients’ characteristics.

The age was higher in the third wave (68 ± 10 years), compared to the first two waves (63 ± 9 and 65 ± 12 years, *p* < 0.001). IHD (18% vs. 8% and 10%, *p* = 0.002) and obesity (25% vs. 15% and 18%, *p* = 0.04) were more prevalent in the third wave, while chronic kidney disease and diabetes were comparable in all three waves. Although stroke was more frequent among patients in wave 3 (namely 9%), we found no clear differences among the three wave groups.

Less than 15% of the patients (37 in total) in wave 3 had received at least one dose of a vaccine (only 27 had received two-dose vaccination and none had received a booster). The received vaccines were Pfizer (26 patients), Moderna (three patients), Astra Zeneca (three patients), and Johnson & Johnson (five patients). No information regarding the time span from vaccination to hospital admission was available.

More than one-third of the patients in all waves required an oxygen mask at least (Table 1). Data retrieved from the hospital electronic database regarding noninvasive vs. invasive ventilation did not generate independent variables; therefore, no further analysis followed this step.

Table 2 shows the outcomes sought in this analysis: in-hospital mortality and disease severity assessed by length of hospitalization and need for mechanical ventilation. Longer hospitalization was significantly associated with wave 3 in comparison to the first and second waves (11 ± 5 vs. 8 ± 4 and 10 ± 5, *p* < 0.001). Mortality rates were significantly higher in the third wave, compared to the first and second waves (32% vs. 13% and 25%, *p* < 0.001). Acute respiratory failure and severe cases of COVID-19, defined by the need for mechanical ventilation, were more frequent in the third wave in comparison with the first two waves (21% vs. 8% and 9%, *p* < 0.001).

### 3.2. Relationship between Primary and Secondary Outcomes

Table 3 presents the secondary outcomes (i.e., length of hospitalization and required mechanical ventilation) in relation to the primary outcome (i.e., in-hospital mortality). Since the relation was highly significant from statistical point of view, and the values in Table 1 proved a highly significant relationship between these two secondary outcomes and the three wave groups, we further investigated the two-way associations (Figure 2). While hospitalization significantly tended to increase over the three wave groups, in case of patients who died, it proved to have a high degree of variability (Figure 2a). In contrast, mechanical ventilation was significantly associated with a very high risk of in-hospital death across all three waves (Figure 2b).

Overall, the primary outcome was significantly dependent on the secondary outcomes.

### 3.3. Relationship between Patients’ Characteristics and the Primary Outcome

Patients who died in the hospital ere about six years older than those discharged alive, a difference that was highly significant (*p* < 0.001). More than half presented with dyspnea, but almost two-thirds of them had a favorable outcome (*p* < 0.001). Thirty-four percent of obese patients died in the hospital (*p* = 0.005). There was a statistically significant association between in-hospital mortality, on the one hand, and IHD and stroke, on the other hand (*p* = 0.015 and *p* = 0.027, respectively). Table 4 synthesizes the relationship between patients’ characteristics and in-hospital mortality.

The presence of IHD was significantly associated with a higher risk of in-hospital death (Table 4) and had an increased prevalence over the three wave groups (Table 1). Considering these findings, we further investigated the association of IHD with in-hospital mortality on the three group strata. Table 5 details the association (including the OR values) between IHD and in-hospital mortality for all patients, on the one hand, and separately for each wave, on the other hand.

Overall, there was a significant association between IHD and in-hospital death; i.e., the patients with no IHD were about 80% more likely to be discharged alive. When we separately considered each group, the association did not reach statistical significance for any of them, but the aggregate OR (the common estimate for risk) was similar, although marginally significant. The Breslow–Day test confirmed the homogeneity of risk across the three wave strata (*p* = 0.387); therefore, IHD might be a medical condition that worsened the odds of patients’ survival irrespective of the three pandemic waves included in this descriptive analysis. The results were similar in cases with required mechanical ventilation.

Figure 3 shows the OR values of IHD patients in relation to in-hospital death and required mechanical ventilation across the three waves.

Similarly, all characteristics and variables in Table 1 and Table 4 (namely those regarding the CVD comorbidities and additional characteristics related to such chronic medical conditions) were investigated to explore their connections with unfavorable outcomes. Only dyspnea and obesity proved to have statistically significant associations with in-hospital death, but the implied risk was inhomogeneous across the three wave strata, as assessed by the Breslow–Day statistical test: *p* = 0.022 and *p* = 0.003 for dyspnea and obesity, respectively.

Figure 4 illustrates the relationship between each of these two characteristics and the primary outcome sought in this investigation. The risk entailed by dyspnea was statistically significant in all three waves, but it significantly decreased from one wave to the next. The risk associated with obesity was statistically significant only in wave 3.

### 3.4. Results of Paraclinical Investigations in Relation to the Primary Outcome

Table 6 details the clinical characteristics and investigations in relation to the primary outcome across the three wave groups. When data were missing in the electronic patient records, the actual number of values used for statistics is specified.

High N/L and T/L ratios were significantly associated with in-hospital death (*p* < 0.001). There was a statistically significant relationship between in-hospital mortality and increased levels of CRP, procalcitonine, or troponine (*p* < 0.001 for each). The fibrinogen levels were significantly higher for patients with unfavorable outcomes in the first two waves (*p* = 0.045 and *p* = 0.036 in waves 1 and 2, respectively). Although an infrequent parameter in the first two waves, increased levels of IL-6 in the third wave were significantly associated with in-hospital mortality (*p* < 0.001).

## 4. Discussion

The three subsets of data concerned the first pandemic waves in Romania, when infection rates were at the highest peaks. This approach was chosen because each wave exhibited distinct characteristics, including: (a) high circulation of different SARS-CoV-2 strains during each specific time period, (b) varying levels of understanding of the disease, resulting in corresponding differences in medical guidelines and standards of care; (c) differences in national and international public health policies; and (d) a gradual development of fatigue among the population with regard to observing imposed rules, accompanied by prolonged deficiencies in medical care for chronic conditions and isolation of the elderly.

The third pandemic wave (corresponding to the delta surge of the virus) was found to be the most aggressive and caused the highest in-hospital mortality for these chronic patients. Increased disease severity, as evidenced by significant associations with longer hospitalization periods, and the need for mechanical ventilation were also higher during the third wave. These findings are consistent with the results reported by other studies. For example, in a retrospective cohort study, Fisman et al. [13] found a 235% risk of ICU admission and 133% risk of death among patients infected with the delta strain, compared to 89% and 51%, respectively, among patients infected with the alpha strain. Zali et al. [14] conducted an observational study of Iranian patients hospitalized for COVID-19 from March to October 2021; the delta wave was associated with a higher risk of death compared to the alpha wave. On the other hand, Florensa et al. [15] showed that epidemiological characteristics in Spain were different compared to our study: delta was more prevalent in younger patients, while the alpha variant was more aggressive than delta, showing higher mortality rates during the alpha wave. In a study by Yao et al. [16], the mortality at peak rates of the four COVID-19 waves were compared across 119 countries according to their economic level. While high income countries experienced a higher death toll in the first wave compared to middle- and low-income countries, in the subsequent waves, the statistics were reversed [16]. According to the gross national income per capita obtained from the World Bank Atlas [17], Romania is an upper middle income country, so a lesser death toll should have been apparent in wave 3 (particularly an opposite trend to the previously observed trend). One possible explanation for this misalignment of our results with the general statistics could be rooted in the low percentage of vaccinated people in Romania (only 30% of the general population at the peak of the third wave), which led to the highest mortality rates and ICU admissions (91% of ICU patients were unvaccinated), according to the Romanian National Government Statistics Data [18]. The national COVID-19 Stringency Index, developed by OxGCRT [19,20], evaluates the strictness of each national government’s policies and public health measures in response to the pandemic progression. With an index of 70 during the third pandemic wave, Romania was similar to other European countries (such as Italy, the United Kingdom, France, or Germany), but the death toll remained high due to the low proportion of vaccination (in western European countries, the percentage of the vaccinated population has varied between 50% and 70%).

We also observed that the prevalence of IHD patients infected with the SARS-CoV-2 virus doubled from 8% in the first wave to 18% in the third wave. Moreover, obesity was more prevalent in the third wave. As the infection rate among patients with IHD continued to escalate with each wave, another observed fact was that IHD was significantly associated with in-hospital death and the need for mechanical ventilation, while patients with no IHD were 80% more likely to be discharged alive. In a meta-analysis of more than 20,000 patients, those with IHD had a three-fold risk of mortality and high COVID-19 disease severity [21]. There have been several studies that pointed out pre-existing cardiovascular comorbidities as predictive of COVID-19 disease severity and mortality [7,22,23]. However, these results have been conflicting with other studies showing no independent association of COVID-19 mortality with CVD in general but rather depending on the type of CVD [8,24].

In all three waves, the following characteristics were associated with increased in-hospital mortality: older age (>70 years), male gender, prolonged hospitalization, the need for mechanical ventilation, dyspnea, IHD, and stroke. There was no change in the association between IHD and mortality according to the three waves; actually, its presence might be considered a medical condition that worsened the odds of patients’ survival irrespective of the three pandemic waves included in this study. We found an obesity-associated risk only in wave 3, but in a large meta-analysis of more than three million COVID-19 patients, obesity was found to be consistently associated with increased severity and higher mortality rates [25]. Although our findings support other studies regarding the impact of old age and male sex on poor COVID-19 outcomes, Phelps et al. reported that pre-existing cardiovascular comorbidities increased the risk of unfavorable outcomes only in female patients [8]. Regarding the symptoms’ burden in this analysis, dyspnea was the most common symptom among patients in the third wave group; it almost tripled compared to the first wave. Moreover, mortality was three times higher in patients who experienced dyspnea, which is mainly indicative of respiratory failure. In a review by Hentsch et al. [26], dyspnea was cited at highly significantly associated with mortality in COVID-19 patients, with more than a four-fold risk (OR = 4.3, *p* < 0.001), but it was not always correlated with disease severity [26].

Acute cardiac injury was considered in patients with high-sensitive troponine levels greater than the 99th percentile of the upper limit (more than 25 pg/mL). Troponine and procalcitonine levels were strongly associated with inflammatory markers (N/L ratio, T/L ratio, and CRP levels), glycemia, and creatinine levels. Additionally, there was a significant correlation between troponine levels and procalcitonine levels. In a study by Xu et al., cardiac injury (defined as an increase in troponine level) was highly associated with increased CRP levels, older age, underlying comorbidities, and COVID-19 disease severity [27]. Systemic inflammatory responses, along with immunometabolism alterations, may lead to myocardial inflammation, with direct myocardial injury resulting in the rise of troponine levels or indirectly by destabilization of plaques in patients with coronary heart disease through microvascular dysfunction and consequent acute coronary syndromes [28,29]. On the other hand, procalcitonine, which is considered a biomarker of systemic bacterial infection (in other words, sepsis), can explain the correlation with troponine levels, as sepsis is responsible for acute myocardial injury [30]. In these CVD patients, the following clinical variables were significantly associated with unfavorable outcomes in all pandemic waves: increased N/L and T/L ratios; high CRP; and increased procalcitonine and troponine levels. Increased serum concentrations of CRP, procalcitonine, and other inflammatory markers have also been reported to be associated with COVID-19 disease severity and poor outcomes by other reports [31,32]. CRP is an acute-phase inflammatory protein produced by the liver in conditions of inflammation and infection. Raised CRP levels were previously demonstrated to be a prognostic marker; however, in the COPE study, CRP levels were lower in the second wave compared to the first wave [33]. In our study, the trend for the CRP level was toward an increase from wave 1 to 3, implying a higher inflammatory response. In a meta-analysis of more than 12,000 patients, elevated cardiac troponine showed 55% sensitivity and 80% specificity for mortality in COVID-19 patients [34]. In a systematic review of more than 13,000 patients, Kumar et al. showed that procalcitonine had good sensitivity and specificity for predicting disease severity and mortality in COVID-19 patients [35]. Based on the current evidence, one can connect hyper-inflammation with disease severity and poor outcomes. On the other hand, the higher values of inflammatory markers that were seen in the third wave might be explained by the delta strain’s virulence. In an experimental study on mice, the delta strain was compared to the alpha, beta, and gamma SARS-CoV-2 strains, and the results showed an enhanced interferon response, high lung tissue inflammation, and cell infiltration with overall increased viral pathogenesis [36].

Overall, the short-term outcomes and cardiac injury in our study were unfavorable for those with a cardiovascular burden, and this outcome was in accordance with other studies in which patients with COVID-19 infection were found to be at high risk for mortality and major cardiovascular events in the acute phase, with the risk extending over the long term; these findings held even in those with mild forms of infection [11,37].

In these cardiovascular patients, the impact of older age, IHD, obesity, and inflammatory markers on poor COVID-19 outcomes, together with propensity matching of these clinical parameters with the third pandemic wave, implies a possible connection; an inefficient vaccination campaign would expose the elderly and high-risk populations to the highly aggressive and virulent delta strain and thus explain the increased mortality rates seen in our study. IHD and obesity are known conditions due to their increased inflammatory status. In addition to the delta strain’s high virulence, these conditions might explain the hyper-inflammation state associated with the overall mortality seen in our data.

### Limitations

This retrospective analysis was a secondary use of medical data collected in a time of heavy disruption in the healthcare system. Consequently, the limited information and data allowed us to only focus on descriptive analysis and estimate the patterns of the clinical and paraclinical picture for these chronic patients, rather than develop predictive regression models of risk. Therefore, we chose not to calculate a hypothetically required sample size (although doing so entailed uncertainty regarding the statistical power) but rather to concentrate on providing accurate descriptors of these chronic patients across particular spells of the COVID-19 pandemic. We nevertheless acknowledge that the confidence intervals for the risk of in-hospital death were large, especially for wave 1, as a result of large standard errors of these estimates. They thus imply high variability and numerous contributing factors to this outcome during wave 1 in particular.

Another limitation is the lack of data on patients’ evolution beyond the hospitalization period. Moreover, the electronic database did not include data on the actual SARS-CoV-2 strain for each clinical case; we could only assume a higher probability of a certain infection based on statistics at the national and international levels. This fact brings additional caveats regarding the possible inferences based on the results, but this shortcoming was partially compensated for by restraining the time period of each data subset.

Despite its limitations, this timely report contributes to the increasing body of evidence regarding the patterns of unfavorable outcomes for chronic cardiovascular patients diagnosed with COVID-19.

## 5. Conclusions

This debriefing analysis of data from the electronic hospital database revealed that chronic cardiovascular patients infected with SARS-CoV-2 during the third wave of the COVID-19 pandemic experienced higher mortality rates and disease severity compared to the first two waves.

The purpose of this retrospective data scrutiny was to examine how these already vulnerable CVD patients were affected in a country with a high national cardiovascular burden. Its primary strength consists of providing concrete information about this specific group of chronic patients, as well as the medical evidence supporting the potential benefits of vaccination for them, a sensitive issue in the Romanian population since COVID-19 might be an emerging cardiovascular risk factor that is here to stay. The lessons learned from this analysis could guide decision-making at all levels with regard to prioritizing limited resources under the constraints of national public health policies. Additionally, the findings would support healthcare providers in helping individual patients to make informed decisions about their care priorities and choices.

We should also see these findings in a broader context, in which infectious disorders are a major burden for patients with chronic cardiovascular disease, being responsible for high rates of decompensation, hospitalization, and death among this population, especially in the cold season. Our retrospective analysis highlighted the morbidity and mortality potential of SARS-CoV-2 itself through its pathological interaction with the cardiovascular system. Pneumococcal and influenza infections are already known for their increased morbidity in the cardiovascular population; therefore, specific preventive measures, such as seasonal vaccination, are strongly recommended by current cardiovascular guidelines. We believe that vaccination against COVID-19 should become a regular seasonal measure, implemented through information campaigns, family doctor appointments, and cardiac physician consultations.

## Figures and Tables

**Figure 1 healthcare-11-01183-f001:**
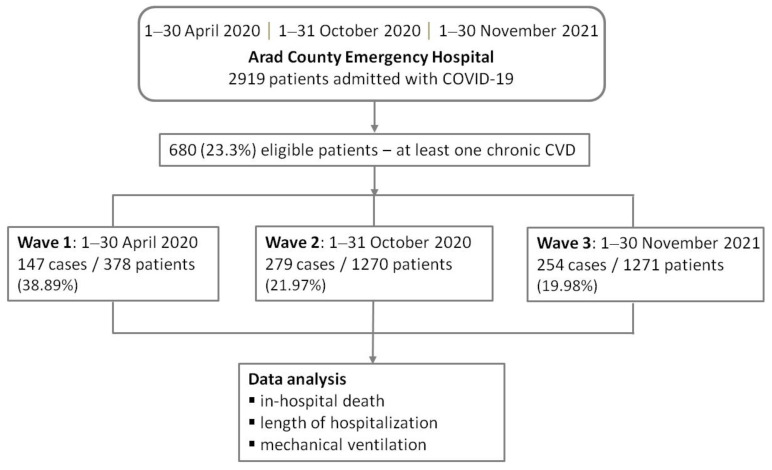
Study flow diagram. Abbreviation: CVD, cardiovascular disease.

**Figure 2 healthcare-11-01183-f002:**
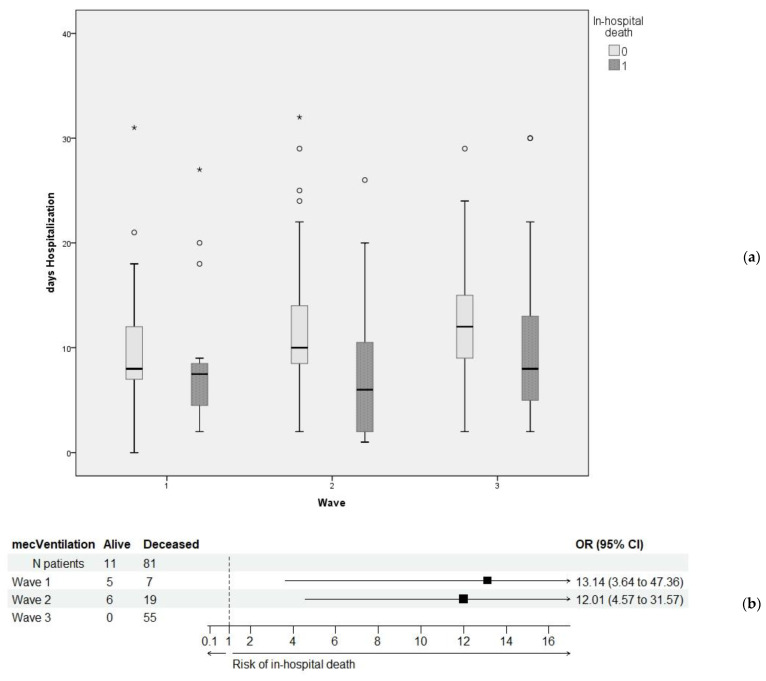
Relationship between the primary and secondary outcomes: (**a**) box-plots for length of hospitalization over the three waves for each possible primary outcome; (**b**) odds ratio (OR) values for in-hospital death when patients underwent mechanical ventilation. In wave 3, all 55 cases who required mechanical ventilation had unfavorable outcomes; therefore, an OR value could not be calculated.

**Figure 3 healthcare-11-01183-f003:**
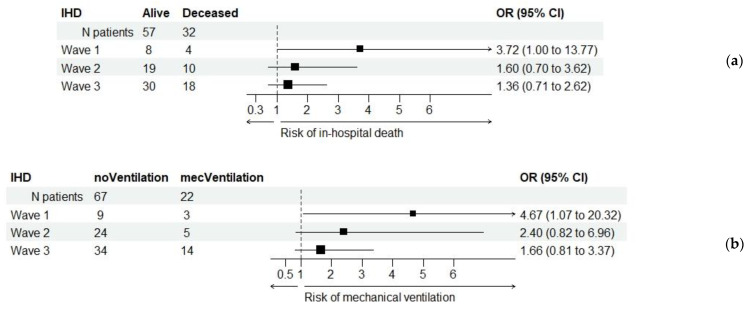
Relationship between ischemic heart disease (IHD) and the primary and secondary outcomes across the three waves: (**a**) odds ratio (OR) values for in-hospital death; (**b**) odds ratio (OR) values for mechanical ventilation.

**Figure 4 healthcare-11-01183-f004:**
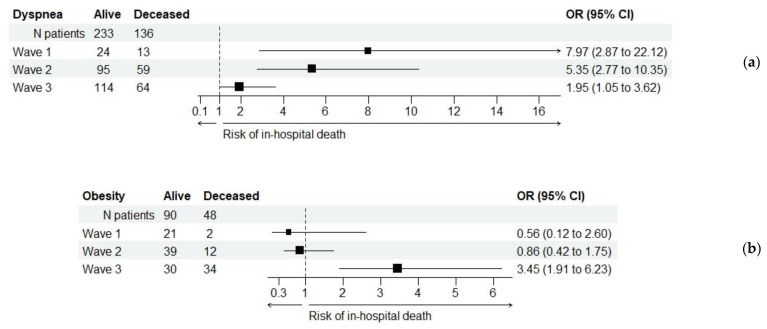
Odds ratio (OR) values for in-hospital deaths in relation to dyspnea (**a**) and obesity (**b**). The risk of in-hospital death associated with dyspnea was significantly decreased in wave 3 (Breslow–Day test, *p* = 0.022), although it remained statistically significant in all three wave groups. The risk associated with obesity was significant only in wave 3.

**Table 1 healthcare-11-01183-t001:** Study participants: descriptive statistics.

Characteristic/ Variable	All Patients (Total 680)	Wave 1 April 2020 (Total 147)	Wave 2 Nov 2020 (Total 279)	Wave 3 Oct 2021 (Total 254)	*p*-Value ^(a), (b)^
Age in years ^(a)^	66.22 ± 11.44	63.18 ± 9.77	65.56 ± 12.72	68.71 ± 10.31	<0.001 **
	67 (59–74)	64 (56.5–70)	67 (57–74)	70 (63–76)	
Sex, M ^(b)^	328 (48.2%)	85 (57.8%)	143 (51.3%)	100 (39.4%)	0.001 **
Dyspnea ^(b)^	369 (54.3%)	37 (25.2%)	154 (55.2%)	178 (70.1%)	<0.001 **
Hypertension ^(b)^	643 (94.5%)	144 (98%)	279 (100%)	220 (86.6%)	<0.001 **
IHD ^(b)^	89 (13.1%)	12 (8.2%)	29 (10.4%)	48 (18.9%)	0.002 *
VHD ^(b)^	16 (2.4%)	2 (1.4%)	2 (0.7%)	12 (4.7%)	0.006 **
Pulmonary embolism ^(b)^	3 (0.4%)	1 (0.7%)	0	2 (0.8%)	0.333
CKD ^(b)^	61 (9%)	10 (6.8%)	23 (8.2%)	28 (11%)	0.311
Stroke ^(b)^	46 (6.8%)	9 (6.1%)	13 (4.7%)	24 (9.4%)	0.084
COPD ^(b)^	44 (6.5%)	18 (12.2%)	26 (9.3%)	0	<0.001 **
DM ^(b)^	245 (36%)	51 (34.7%)	100 (35.8%)	94 (37%)	0.894
Obesity ^(b)^	138 (20.3%)	23 (15.6%)	51 (18.3%)	64 (25.2%)	0.040 *
Score XR ^(a)^	2.58 ± 2.56	2.22 ± 2.13	3.21 ± 2.58	2.11 ± 2.62	<0.001 **
	2 (0–4)	2 (0–4)	3 (0–6)	0 (0–4)	
Vaccinated ^(b)^	37 (5.4%)	–	–	37 (14.6%)	–
O_2_ simple mask ^(b)^	277 (40.7%)	51 (34.7%)	90 (32.3%)	136 (53.5%)	<0.001 **
NIV ^(b)^	185 (27.2%)	51 (34.7%)	130 (46.6%)	4 (1.6%)	<0.001 **
OTI ^(b)^	94 (13.8%)	12 (8.2%)	25 (9%)	57 (22.4%)	<0.001 **

^(a)^ mean ± standard deviation; median (interquartile Range), with Tukey’s hinges; Kruskal–Wallis-test;

^(b)^ observed frequency (percentage); chi-square test (either asymptotic, Fisher’s exact test, or Monte-Carlo simulation with 10,000 samples, as appropriate);

Statistical significance: *, *p* < 0.05; **, *p* < 0.01;

Abbreviations: CKD, chronic kidney disease; COPD, chronic obstructive pulmonary disease; DM, diabetes mellitus; IHD, ischemic heart disease; NIV, noninvasive ventilation; OTI, orotracheal in-tubation; VHD, valvular heart disease; XR, X-ray radiography.

**Table 2 healthcare-11-01183-t002:** Study outcomes across the three pandemic waves: descriptive statistics.

Characteristic/ Variable	All Patients (Total 680)	Wave 1 April 2020 (Total 147)	Wave 2 Nov 2020 (Total 279)	Wave 3 Oct 2021 (Total 254)	*p*-Value ^(a), (b)^
Death ^(a)^	173 (25.4%)	20 (13.6%)	72 (25.8%)	81 (31.9%)	<0.001 **
Days hosp ^(b)^	10.44 ± 5.19	8.94 ± 4.74	10.19 ± 5.06	11.58 ± 5.34	<0.001 **
	10 (8–13)	8 (7–11)	10 (8–13)	11 (8–15)	
Mechanical ventilation ^(a)^	92 (13.5%)	12 (8.2%)	25 (9%)	55 (21.7%)	<0.001 **

^(a)^ observed frequency (percentage); chi-square test;

^(b)^ mean ± standard deviation; median (interquartile range), with Tukey’s hinges; Kruskal–Wallis-test;

Statistical significance: **, *p* < 0.01.

**Table 3 healthcare-11-01183-t003:** Length of hospitalization and mechanical ventilation in relation to in-hospital mortality.

Characteristic/ Variable	All Patients (Total 680)	Discharged Alive (Total 507)	In-Hospital Death (Total 173)	*p*-Value ^(a), (b)^
Days hosp ^(a)^	10.44 ± 5.19	11.16 ± 4.64	8.32 ± 6.08	<0.001 **
	10 (8–13)	10 (8–14)	7 (3–11)	
Mechanical ventilation ^(b), (c)^	92 (13.5%)	11 (12%)	81 (88%)	<0.001 **

^(a)^ mean ± standard deviation; median (interquartile range), with Tukey’s hinges; Mann–Whitney U-test;

^(b)^ observed frequency (percentage); chi-square test;

^(c)^ the percentages for column two were calculated out of the sample total; the percentages for columns three and four were separately calculated for each row total, so they total 100% for that particular row; the chi-square test was applied for the significance of the association between the presence/absence of a condition or symptom and in-hospital mortality (i.e., not for differences in the proportion of in-hospital deaths/survival when the condition is present);

statistical significance: **, *p* < 0.01.

**Table 4 healthcare-11-01183-t004:** Relationship between patients’ characteristics and in-hospital mortality.

Characteristic/ Variable	All Patients (Total 680)	Discharged Alive (Total 507)	In-Hospital Death (Total 173)	*p*-Value ^(a), (b)^
Age in years ^(a)^	66.22 ± 11.44	64.73 ± 11.45	70.59 ± 10.26	<0.001 **
	67 (59–74)	66 (57–73)	71 (64–77)	
Sex M ^(b), (c)^	328 (48.2%)	238 (72.6%)	90 (27.4%)	0.036 *
Sex F ^(b), (c)^	352 (51.8%)	269 (76.4%)	83 (23.6%)	
Dyspnea ^(b), (c)^	369 (54.3%)	233 (63.1%)	136 (36.9%)	<0.001 **
Hypertension ^(b), (c)^	643 (94.6%)	486 (95.9%)	157 (90.8%)	0.011 *
IHD ^(b), (c)^	89 (13.1%)	57 (64%)	32 (36%)	0.015 *
VHD ^(b), (c)^	16 (2.4%)	9 (56.2%)	7 (43.8%)	0.141
Pulmonary embolism ^(b), (c)^	3 (0.4%)	1 (33.3%)	3 (66.7%)	0.161
CKD ^(b), (c)^	61 (9%)	40 (65.6%)	21 (34.4%)	0.091 ^#^
Stroke^(b), (c)^	46 (6.8%)	28 (60.9%)	18 (39.1%)	0.027 *
COPD ^(b), (c)^	44 (6.5%)	36 (81.8%)	8 (18.2%)	0.253
DM ^(b), (c)^	245 (36%)	175 (71.4%)	70 (28.6%)	0.160
Obesity ^(b), (c)^	138 (20.3%)	90 (65.2%)	48 (34.8%)	0.005 **
Score XR ^(a)^	2.58 ± 2.56	2.44 ± 2.38	3.01 ± 2.98	<0.106
	2 (0–4)	2 (0–4)	2 (0–6)	
Vaccinated ^(b)^	37 (5.4%)	26 (70.3%)	11 (29.7%)	0.76
O2 simple mask ^(b)^	277 (40.7%)	231 (83.4%)	46 (16.6%)	<0.001 **
NIV ^(b)^	185 (27.2%)	166 (89.7%)	19 (10.3%)	<0.001 **
OTI ^(b)^	94 (13.8%)	39 (41.5%)	55 (58.5%)	<0.001 **

^(a)^ mean ± standard deviation; median (interquartile range), with Tukey’s hinges; Mann–Whitney U-test;

^(b)^ observed frequency (percentage); chi-square test (either asymptotic, Fisher’s exact test, or Monte-Carlo simulation with 10,000 samples, as appropriate);

^(c)^ the percentages for column two were calculated out of the sample total; the percentages for columns three and four were separately calculated for each row total, so they total 100% for that particular row; the chi-square test was applied for the significance of association between the presence/absence of a condition or symptom and the in-hospital mortality (i.e., not for differences in the proportion of in-hospital deaths/survival when the condition was present);

Statistical significance: ^#^, *p* < 0.1; * *p* < 0.05; ** *p* < 0.01;

Abbreviations: CKD, chronic kidney disease; COPD, chronic obstructive pulmonary disease; DM, diabetes mellitus; IHD, ischemic heart disease; NIV, noninvasive ventilation; OTI, orotracheal in-tubation; VHD, valvular heart disease; XR, X-ray radiography.

**Table 5 healthcare-11-01183-t005:** Association of ischemic heart disease (IHD) with in-hospital mortality on the three-wave strata.

Ischemic Heart Disease		In-Hospital Death			*p*-Value	OR (95% CI)
		–	+	Total		
All patients	–	450	141	591	0.015 *	1.792 (1.117; 2.874)
	+	57	32	89		
Wave 1	–	119	16	135	0.060	3.719 (1.005; 13.766)
	+	8	4	12		
Wave 2	–	188	62	250	0.259	1.596 (0.704; 3.615)
	+	19	10	29		
Wave 3	–	143	63	206	0.354	1.362 (0.707; 2.622)
	+	30	18	48		
Breslow–Day test: chi-square = 1.896 (2 df), *p* = 0.387						NOTE for separate strata analysis: statistically non-significant difference between the strata and a marginally significant overall OR value
Mantel–Haenszel test: chi-square = 3.387 (1 df), *p* = 0.066						
Mantel–Haenszel common estimate for risk OR = 1.604, 95% CI (0.996; 2.586)						

Statistical significance: * *p* < 0.05;

Abbreviations: CI, confidence interval; OR, odds ratio.

**Table 6 healthcare-11-01183-t006:** Investigations’ results in relation to in-hospital mortality across the three-wave groups.

Variable	Wave	All Patients	Discharged Alive	In-Hospital Deaths	*p*-Value ^(a)^
N/L ratio ^(a)^	1	n = 146	n = 127	n = 19	
		3.09 (1.87–6.49)	2.79 (1.71–4.59)	13.42 (7.37–18.68)	<0.001 **
	2	n = 270	n = 198	n = 72	
		5.845 (3.00–11.53)	4.90 (2.83–8.43)	12.03 (5.19–19.29)	<0.001 **
	3	n = 254	n = 173	n = 81	
		5.92 (3.86–9.85)	5.1 (3.33–7.65)	7.92 (5.41–15.69)	<0.001 **
T/L ratio ^(a)^	1	n = 146	n = 127	n = 19	
		9.175 (6.07–18.11)	8.14 (5.75–14.33)	29.30 (17.03–59.64)	<0.001 **
	2	n = 270	n = 198	n = 72	
		15.26 (7.91–31.56)	13.41 (7.48–26.04)	25.14 (9.69–52.09)	< 0.001**
	3	n = 254	n = 173	n = 81	
		17.22 (9.78–31.12)	14.21 (8.76–25.67)	22.53 (13.65–44.62)	<0.001 **
Fibrinogen ^(a)^	1	n = 147	n = 127	n = 20	
		501 (387.5–625)	500 (380.5–597)	599.5 (434–718.5)	0.045 *
	2	n = 182	n = 146	n = 36	
		585 (452–704)	564.5 (445–699)	657 (502.5–744.5)	0.036 *
	3	n = 247	n = 166	n = 81	
		540 (439–651)	540 (436–646)	534 (451–670)	0.827
CRP ^(a)^	1	n = 136	n = 119	n = 17	
		24.71 (5.395–76.66)	18.35 (4.955–48.495)	104.82 (83.07–155.64)	<0.001 **
	2	n = 165	n = 130	n = 35	
		39.9 (8.84–95.06)	27.885 (7.93–71.88)	85.27 (37.29–188.815)	<0.001 **
	3	n = 240	n = 164	n = 76	
		86.84 (40.735–139.44)	77.37 (33.73–130.955)	105.515 (57.06–163.2)	0.002 **
Procalcitonine ^(a)^	1	n = 146	n = 126	n = 20	
		0.0645 (0.048–0.168)	0.06 (0.047–0.109)	0.526 (0.26–1.875)	<0.001 **
	2	n = 111	n = 93	n = 18	
		0.10 (0.06–0.22)	0.09 (0.05–0.17)	0.32 (0.15–1.83)	<0.001 **
	3	n = 209	n = 146	n = 63	
		0.12 (0.06–0.25)	0.09 (0.06–0.17)	0.24 (0.13–1.18)	<0.001 **
Troponine ^(a)^	1	n = 52	n = 39	n = 13	
		11.2 (7.38–27.735)	9.64 (6.74–12.93)	38 (29–107)	<0.001 **
	2	n = 29	n = 22	n = 7	
		8.6 (5.04–21.9)	6.935 (4.7–12.05)	41.75 (28.405–73.51)	0.001 **
	3	n = 66	n = 32	n = 34	
		16.75 (10.9–36.7)	12.8 (9.15–19.3)	25.25 (13.7–102.7)	<0.001 **
IL-6 ^(a)^	1	n = 6	n = 6	–	
		6.45 (2.44–7.07)	6.45 (2.44–7.07)	–	–
	2	–	–	–	–
	3	n = 207	n = 138	n = 69	
		50.05 (18.4–101.15)	36.96 (11.99–74.35)	81.9 (32.38–173)	<0.001 **

^(a)^ mean ± standard deviation; median (interquartile range), with Tukey’s hinges; Mann–Whitney U-test for each wave;

statistical significance: *, *p* < 0.05; **, *p* < 0.01;

Abbreviations: CRP, C reactive protein; IL-6, interleukin-6; N/L, neutrophil/leukocyte ratio, T/L, thrombocyte/leukocyte ratio.

## Data Availability

Raw data were retrieved from the patients’ records and are available only upon formal request and approval of the hospital management and ethics committee.

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
