# Peer review of "Successive Waves of the COVID-19 Pandemic Had an Increasing Impact on Chronic Cardiovascular Patients in a Western Region of Romania"

_healthcare, 2023, doi:10.3390/healthcare11081183_

Round 1
Reviewer 1 Report
Thank you for your effort in creating the paper entitled "Successive Waves of the COVID-19 Pandemic Had an Increasing Impact on Chronic Cardiovascular Patients in a Western Region of Romania". I have some remarks:
- The Introduction is very well written and contains all the needed information. Congratulations! I found a slight spelling/punctuation error in line 78. Please check and correct.
- The information given in the last two paragraphs of the Introduction is repeated unnecessarily in the Methods. Please rewrite.
- You gave the sum of the patients with positive COVID in the study. Please add the information on how many patients out of all included (n, %) had CVD in each wave; this might be included in your Figure 1. It is very nice.
- In table 1, you mix the description of your population with your primary/secondary outcomes (death, length of stay, mech ventilation). Please correct.
- It would be of great value if you included the results concerning noninvasive ventilation use (Bipap, HFNC) in your patients.
- Please define what do you considered IHD in your study,
- Please add the information about the number of vaccinated patients in groups.
Reviewer 2 Report
Dear Authors,
I think that your paper is a good work. You could improve the fluency of the text.
I think that the paper has the limitation of being a retrospective analysis. Moreover the analysis is made on a relatively small number of patients, considered the disease and the registry used. However the statistical analysis is well done and I think that the paper could describe a different behavior of COVID 19 in a middle income country.
Best Regards
Reviewer 3 Report
There has been numerous studies on this topic so novelty may not be there. However this is specific to Romania which has lower vaccinated individuals and therefore we do see different outcomes to other studies.
I note very large confidence intervals in their results due to lower power specially when comparing in hospital deaths in the three waves.
Overall a well written and conducted study however may be a bit out of date at this point in time.
